# Neonatal Care Unit Interventions on Preterm Development

**DOI:** 10.3390/children10060999

**Published:** 2023-06-02

**Authors:** Alexia Séassau, Pascale Munos, Catherine Gire, Barthélémy Tosello, Isabelle Carchon

**Affiliations:** 1Centre Hospitalier du Pays d’Aix, 13100 Aix-en-Provence, France; 2Hôpital Nord de Marseille, 13015 Marseille, France; 3Department of Neonatology, Hôpital Nord de Marseille, 13015 Marseille, France; 4CHART Human and Artificial Cognition Laboratory at Ephe, École Pratique des Hautes Études-PSL Paris-Sciences-Lettres, 93322 Aubervilliers, France

**Keywords:** prematurity, incubator, uni- and multi-sensorial solicitations, parents

## Abstract

Prematurity is becoming a real public health issue as more and more children are being born prematurely, alongside a higher prevalence of neurodevelopmental disorders. Early intervention programs in Neonatal Intensive Care Units (NICUs) correspond to these uni- or multi-sensorial solicitations aiming to prevent and detect complications in order to support the development of preterm infants. This article aims to distinguish sensory intervention programs according to the gradient of the type of solicitations, uni- or multi-modal, and according to the function of the person who performs these interventions. Uni-sensorial interventions are essentially based on proprioceptive, gustatory, or odorant solicitations. They allow, in particular, a reduction of apneas that support the vegetative states of the preterm infant. On the other hand, the benefits of multi-sensory interventions seem to have a longer-term impact. Most of them allow the support of the transition from passive to active feeding, an increase in weight, and the improvement of sleep-wake cycles. These solicitations are often practiced by caregivers, but the intervention of parents appears optimal since they are the main co-regulators of their preterm child’s needs. Thus, it is necessary to co-construct and train the parents in this neonatal care.

## 1. Introduction

Each year, it is estimated that 1 in 10 births worldwide, or 15 million children, are born prematurely [1]. Medical progress in perinatal care since the year 2000 has allowed the survival of these increasingly immature preterm children with low birth weights, resulting in an increasingly long hospitalization period [2]. 

Children born prematurely have sensory and perceptual peculiarities, with atypicalities depending on their experience in utero, their age, their birth weight, and the number of days spent in the neonatology department [3]. These atypicalities in processing information from the external environment alter their sensory-motor exploration abilities [4,5]. A higher prevalence of neurodevelopmental deficits exists in these preterm infants compared to full-term newborns, notably concerning intellectual and executive functions, attention, language, and social cognition [5,6,7,8]. The consequences of prematurity are therefore becoming a real public health issue [9].

Early interventions in Neonatal Intensive Care Units (NICUs) seek to limit deviant developmental trajectories. Since the 1980s, the introduction of early, individualized developmental care has shifted the focus from child survival to supporting the well-being of the child and the parents. In the incubator, the preterm newborn is exposed to inappropriate stimulation while in a critical period of physiological immaturity and cerebral development associated with atypical sensoriality. The technical environment of the NICU and the developmental fragility of preterm infant lead to disturbed parent–child interactions [10]. Consequently, the study of parent–child interactions is supported and included in the care process of these prematurely born children [11,12]. 

This work examines both the incubator environments and the type of multi-sensory solicitations most favorable to the sensoriality of preterm infant while questioning the place of caregivers and parents. 

## 2. The Sensory Environment of NICU Incubators and Its Consequences on the Preterm Infant 

Neonatal Intensive Care Units (NICUs) for very preterm infant do not provide the same living and development conditions as the intrauterine environment. preterm infant in incubators are cut off from the prenatal period; in the first moments of their aerial life, they are at odds with the normal biological continuum and immersed in an environment far removed from the intrauterine sensory world [13]. 

During their first days of life, these preterm infants will be confronted with numerous stress factors, such as painful care procedures and frequent and uncomfortable manipulations [13]. The preterm infant in an incubator is exposed to early sensory experiences, atypical in quantity and quality and inappropriate to his or her level of sensory maturation, whereas the temporal sequence of sensory development in utero during the 3rd trimester of gestation is well known: somesthetic and deep tactile sensitivity (proprioceptive) and chemosensory sensitivity (gustation and olfaction) at 14 weeks, vestibular sensitivity (body movement and balance) at 25 weeks, auditory sensitivity at 26 weeks, and then visual sensitivity at 28 weeks [14]. Indeed, fetal sensory experience prepares the organism to interact with the sensory environment after birth. Early mature vestibular and tactile receptors are understimulated during an incubator stay, with isolation and reduced wear time outside of skin-to-skin [14]. While immature sensory modalities in the preterm infant, such as hearing and vision, are conversely over-stimulated [15,16,17,18], including machine noises and bells and distorted voices that are amplified by the incubator walls. Early auditory experiences affect brain development; along with the noise environment of the ward, noise can be associated with tachycardia or bradycardia, apneas, decreased oxygenation, increased muscle tension, blood pressure and intracranial pressure, and sleep disturbances [19]. The noise environment exacerbates the child’s energy expenditure, induces physiological instability, and may affect hearing quality. Indeed, a stay of more than four days in the NICU is a risk factor for hearing loss [20]. 

Much of the physical contact (technical gestures and nursing care) is not soothing and is associated with mostly unpleasant and unfamiliar odors such as disinfectant [21,22,23,24]. Some painful procedures (blood work, catheter placement, fundus, etc.), when not balanced by sufficient exposure to positive tactile experiences, contribute to an attenuation of cortical processing of (non-harmful) tactile stimuli at discharge [25]. The incubator stay combines sensory deprivation, over-stimulation, and/or harmful, uncomfortable, or inappropriate stimulation with direct consequences on the brain maturation of the preterm newborn. Certain syndromes inherent to the health of the preterm infant, such as respiratory distress or ulcerative colitis, are the source of intense stress for both the preterm infant and the parents. All of these painful and dystimulating events are considered to be major stress factors with significant consequences on the child’s state of health, on his or her development, and on brain maturation [18,26]. Indeed, stress responses lead to greater oxygen consumption, notably through the acceleration of the heart rate [26], to the detriment of other functions such as tissue development. It is also noted that the vasoconstriction inherent in the stress response can increase intracranial pressure, associated with hypoxia, favoring the development of intraventricular hemorrhages [27].

### 2.1. Reduction of Sensory Stimuli 

Recognition that sensory stimulation can overwhelm preterm infants and increase physiological signs of stress has led to attempts to reduce sensory input [18]. Because preterm infants have difficulty regulating their homeostasis and responding appropriately to environmental stimuli, this interferes with the development of their perceptual and self-regulatory abilities [28,29,30]. The theoretical approach of the Newborn Individualized Developmental Care Assessment Program (NIDCAP) is that the amount of stimulation that preterm babies receive is excessive [31,32]. NIDCAP is a set of strategies designed to protect the preterm infant [33]. They include environmental modifications to minimize stressful sensory stimuli for preterm infants, such as reducing bright lights, reducing loud and sudden noises, limiting temperature changes, and careful observation of the child and his or her ability to regulate care, in order to provide an individualized care program based on each child’s resources and weaknesses [31,33,34]. The NIDCAP approach takes into account the organization of four functioning systems in relation to the external environment: the autonomic system, the motor system, the arousal system, and the interactional system [33,34]. The disorganization of one of these systems as a result of stress has repercussions on the others. The positive effects of NIDCAP can be seen in the short term on weight-bearing growth, respiratory autonomy, and decrease in the length of hospitalization [28,35,36]. NIDCAP enhanced neurobehavioral and neurological development in preterm infants at two weeks of Corrected Age (CA) when compared with standard care [37]. The benefits of NIDCAP were evident at 9 months of age but did not persist at 12, 18, or 24 months of CA [36,38,39,40]. The NIDCAP intervention reduces the need for respiratory support and the length of hospital stay [18]. NIDCAP is a program that is not readily available to all NICUs because it requires extensive training and a significant time investment by caregivers [41].

Some other early intervention programs, such as that of Becker et al. [29], aim to facilitate the preterm infant’s self-regulatory abilities through sound reduction. Decreased noise in the unit and incubator reduces apneas, hypoxemia, and cardiac accelerations in low-weight, very preterm infants [42]. Reducing sound and light for 12 h at night resulted in improved weight gain and increased sleep time [43]. Noise reduction and consolidation of care to allow for longer sleep time and stress monitoring resulted in better staturo-weight gain, shorter length of hospital stay, and better performance on the Neonatal Behavioral Assessment Scale, particularly on the reflex and self-regulation scales [29]. One somewhat contradictory study found no change in physiologic parameters when auditory input was reduced by placing earmuffs on preterm infants for 2 days [44]. Autcott et al. [18] questioned what level and duration of sound exposure is harmful to the developing auditory system of a preterm infant. In animal studies, including gerbils (small rodents), Caras and Sanes [45] and Ihlefeld et al. [46] were able to test the extent to which a disturbance occurring at the very beginning of auditory function later alters behavioral performance by temporarily and partially depriving them of auditory inputs for a brief period, 12 days starting 11 days after birth (by earplugs attenuating airborne sounds by 40 dB SPL, without suppressing bone conduction and the possibility for the animal to hear its own biological sounds). Gerbils showed atypical sensory perception with impaired discrimination of sound localization [44] as well as impaired discrimination of sounds in a noisy environment [46]. In relation to the developmental difficulties of the preterm child, Mowery et al. [47] have shown that there are interactions between the critical period of development of the visual system and that of the auditory system: inducing the visual system to function prematurely disrupts the critical period of development of the auditory system; conversely, delaying the onset of function of the visual system can extend the critical period of the auditory system by several days. These data suggest that if the natural sequence of sensory function development is disrupted, as it is in very preterm infants, the critical period of auditory developmental plasticity may be altered and could lead to alterations [48].

### 2.2. Enrichment of Sensory Stimuli in the NICU

Other programs, on the other hand, offer soothing sensory enrichment in the NICU such as therapeutic touch, soft music, etc. [18]. These programs are based on the underlying theoretical premise that preterm infants suffer from sensory deprivation that limits their development. These stimuli must be provided while taking into account the sequential development of the sensory system [49,50,51] in order to compensate for the dystimulations of the neonatal environment. 

Several sensory enrichment programs have measured improved cognitive development in infants who received these sensory stimuli. Scarr-Salapateck and Williams [52] offered babies in the NICU a combination of visual, tactile, auditory, and kinesthetic stimulation and repeated home visits upon discharge to continue early support. In the intervention group of this study, the preterm infant showed better cognitive and social performance at four and twelve months compared to the control group. 

Introducing a soothing solicitation such as lullabies sung by the parent during a skin-to-skin session induces a better quality of parent–child interactions from the first session [53]. Indeed, singing stabilizes the mother’s gaze on her child with longer fixation times and favors the preterm child’s state of relaxation, with more time with eyes closed compared to skin-to-skin sessions without lullabies. Singing a lullaby during a skin-to-skin session helps to create a better synchronization of rhythms between the mother and her preterm baby [53]. 

The Supporting and Enhancing Sensory Experiences (SENSE) program [54] has studied the effect of positive, age-appropriate sensory input to the preterm infant. These interventions, performed by parents or a team of caregivers when parents are not available, are provided daily during hospitalization. These solicitations include massage, sound enrichment (human voice and music), olfactory enrichment with scented fabric, and vestibular and visual enrichment with dimmed lights [55,56]. Preterm infants who received the SENSE program had less asymmetry on the NeoNatal Neurobehavioral Scale NNNS and higher scores on the Hammersmith Neonatal Neurological Assessment [57], and mothers showed more confidence.

### 2.3. Between “Too Much and Not Enough”, the Right Balance in the NICU

It may seem contradictory that programs aiming at either a reduction of stimuli or an increase in stimuli have favorable consequences for the preterm child [58]. Studies show positive results with both increased and reduced stimuli on short-term physiological measures (including maturation of the alertness system), on neonatal neurobehavioral aspects, or on long-term measures of higher cognitive function [49,59]. Some programs will specifically target this state of alertness with an improvement in the organization of preterm states. Stimulatory programs induce an increase in wakefulness states, whereas those that aim to reduce demands result in an increase in sleep. More broadly, Horowitz [49] suggests that all types of interventions that respect the level of sensory maturation promote homeostasis in the preterm infant. Feldman [50] describes that too much harmful stimulation is detrimental, but too little stimulation can be detrimental, hence the term “too much and not enough”.

Existing interventions based on opposing theoretical underpinnings (inability of preterm children to assimilate multiple sensory information or, conversely, deprivation of sensory stimuli they should have received in utero) have led to opposing recommendations [41], which each have their limitations: these programs are limited to the time of the intervention and the length of hospitalization in the wards.

Whatever the theoretical basis adopted, the aim of these interventions is not to provide a preterm child with experiences that he or she has missed, but to help him or her to extract information from his or her environment by organizing active cycles of receiving adapted stimulation and rest [60]. 

We must try to take into account all of the recommendations and ensure that they are applied over a longer period of time while not overloading the immature neurological system of the preterm infant [61]. In order to promote early development, the consensus, which may seem obvious, suggests providing the preterm infant with stimuli that are neither “too much nor too little” but instead between lack of stimulation and over-simulation by adapting to these sensory needs. Thus, when certain external stimuli are attenuated, such as sounds, vibrations, and lights, and when familiar and pleasant touches and smells are favored, preterm infant can achieve greater well-being [50].

## 3. Early Sensory Intervention Programs in the NICU

Another way to distinguish intervention programs is to focus on the gradient of the type of stimulation, uni- or multi-modal, and/or the function of the person performing the intervention(s), the caregiver or the parent. The objectives of these programs are always to compensate for inappropriate contexts of early development by creating a developmental niche [62,63,64,65].

### 3.1. Uni-Modal Sensory Input

Early intervention programs focusing on uni-modal sensory input have demonstrated specific benefits on neurobehavioral functions in the preterm infant [4,25,50,66].

Programs supporting proprioceptive development and rhythmicity have used hydraulic mattresses that reduce apnea and improve sleep in preterm infants [60]. The use of a stuffed animal called a “breathing bear” on which the preterm infant is placed, and which induces a breathing rhythm, improves the organization of the preterm infant’s sleep and wake states [67]. The repetitive and rhythmic component in the intervention programs, such as hydraulic mattresses, caresses, rocking, “breathing bears”, and massages, tends to increase the interest of the preterm newborn in his environment by providing “external regulators” that allow him to respond in an adequate way to internal and external stimuli [49]. 

Massage therapy is a program based on tactile stimulation of the preterm infant [68,69]. Data have shown an increase in the frequency of active states of arousal during a two-hour observation following ten days of massage therapy [70]. 

The positioning of preterm infants in the incubator provides proprioceptive sensory enrichment that can have major effects on sensorimotor development. Preterm infants placed alternately in symmetrical (e.g., on their backs) and asymmetrical (e.g., on their backs, with the pelvis lifted to one side) positions showed less asymmetry (as measured by the NeoNatal Neurobehavioral Scale) at discharge from the NICU than infants placed exclusively in symmetrical positions [71].

To palliate painful procedures, oral glucose administration has been recommended, resulting in a decrease in facial and vocal expressions of pain. Nevertheless, several studies have questioned the analgesic properties of sweet taste and warn about the long-term effects of cumulative glucose exposure [3]. Indeed, it does not attenuate either pain-specific brain activity or the infant’s spinal nociceptive reflex. While sweet taste can also be used as an attractive stimulus for stimulating food intake, repeated use of glucose could result in aversive conditioning to sweet taste associated with a painful response. Exposure to the odor of breast milk activates mouth movements, has an analgesic effect comparable to that of sugar solution, and decreases apneas [72]. The soothing effects of maternal odor in preterm infants have been shown to reduce crying and decrease stress [72,73]. Recently, melatonin (neurohormone), with its anti-inflammatory functions, has been used in pain control for ventilated preterm infants [74].

Early intervention programs that focus on a single sensory modality also have their limitations. Being centered on a predominant modality, it can sometimes induce over-stimulation, and stress can then result. It seems more natural to stimulate all the senses. More generally, these uni-sensorial stimulations cannot replace social interactions, especially those of the parents.

### 3.2. Multi-Sensory Stimulation 

Multi-sensory solicitation programs, using a combination of visual, auditory, and tactile (and/or kinesthetic) stimuli, also aim to stimulate the neurobehavioral development of preterm infant. This sensory redundancy (or intersensory redundancy) is defined when information is perceived by at least two temporally synchronized sensory modalities [53] and appears to be essential for building a coherent and unified perception of the world, which is the basis for learning and for the synchrony of social interactions [66].

In the 1980s, the first multi-sensory interventions appeared in neonatal services in Bolivia with the Kangaroo Maternal Care method or skin-to-skin care [31,33] The added value of these multi-sensory interventions lies both in their compatibility with Gottlieb’s theory in 1976 [75] on the concept of sequential development of sensory systems and on the primordial place of the mother at the center of this process [76]. This multi-sensory care can be provided by the caregiver, by the parents, or by both.

#### 3.2.1. The Benefits for Preterm Infant of Multi-Sensory Input from Staff 

Multi-sensory care was introduced in neonatal units with a pioneering intervention by caregivers to evaluate the contribution of individualized early care [31,33,77]. NIDCAP-trained caregivers provided individualized care to preterm infants weighing less than 1250 g and less than 28 weeks of age. These high-risk preterm infants who received individualized early care showed a shorter duration of parenteral feeding (intravenous feeding), a faster transition to full oral feeding, a lower incidence of necrotizing enterocolitis (damage to the inner surface of the intestine), a reduction in discharge age, improved weight, and less fussing, startling, and wobbling, resulting in better self-regulation; positive results are also found for parents, since the authors observe a reduction in family stress. 

The systematic review by Cha et al. [78], which examined the effects of interventions on high-risk preterm infants in NICUs, shows that the various types of care provided by nurses help to support the growth and development of these preterm infants. This care is very diverse, such as kangaroo care, massage therapy, listening to the mother’s voice, song or heartbeat, olfactory stimulation through breast milk, or the Yakson technique (Korean therapeutic touch administered to newborns by caressing their abdomen with one hand while the other hand is placed on their back to relieve pain or calm them).

Another study compared four groups of preterm infant whose intervention, performed by nurses, focused solely on physical activity demands [79]. The first group received daily passive exercises of varying amplitude of the upper and lower limbs. The second “hydrotherapy” group received passive, in-water exercises for the shoulders and pelvic area every other day. A combined group received physical activity programs in and out of the water every other day. Preterm infants in the fourth (control) group were maintained in the fetal position with no further intervention. The duration of the intervention was 2 weeks, from 32 to 33 weeks. Test of Infant Motor Performance [80] and neuromuscular scores increased over the 2 weeks, but with no difference between the 4 groups at 34 weeks (except leg recoil, which was significantly higher in the physical activity groups). This care, including NIDCAP, performed by staff has multiple short-term benefits on the development of the preterm infant.

#### 3.2.2. The Benefits of Multi-Sensory Care by the Caregiver and by the Parents 

In the literature, it is sometimes difficult to know what place is offered to the parent in certain care programs [41]: parent as a full actor or parent as a support to the caregiver (with or without the use of his/her voice or smell). Thus, multi-sensory stimulation sessions were proposed to preterm infant with the listening of a recording of the maternal voice combined with her smell (via an impregnated cloth) and with a carrying of the caregiver [66]. Preterm newborns receiving this multi-sensory care added to standard care have, at the exit of the neonatal unit, better sensory adaptation and motor skills as well as more engaged interactions towards the parent with less withdrawal movements, a more relaxed face, and a gaze more turned towards the parent’s voice.

A specific program was developed—Maternal Participation Program (MPP)—integrating the three components of psychosocial support, education, and active participation of parents in the care provided by the nurse [81]. This MPP program is based on the Neonatal Integrative Developmental Care (IDC) model [82], taking into account the authors’ Thai culture. MPP resulted in increased growth (weight and height) and neurobehavioral development of preterm infants in the group whose mothers received MPP with usual nursing care compared with the group whose mothers received only nursing care. This difference was not found at 14 and 28 weeks CA of the preterm infant. Mothers’ participation in care helped them to become involved in the NICU. 

In this sense, Yu et al. [83] clearly demonstrated the prominent and qualitative role of parents in the implementation of care programs in very low birth weight preterm infants: when interactive parent–child dyad activities with supportive classes were offered in addition to standard care interventions by caregivers, earlier discharge, better weight gain with complete enteral feeding, and better neurobehavioral performance were observed compared to the group of preterm infants who received only usual care. The additional role of the parents in standard care appears beneficial for the preterm infant.

#### 3.2.3. The Benefits of Multi-Sensory Solicitations by the Parent

The proprioceptive and rhythmic contact of skin-to-skin provides an experience that minimizes visual and auditory stimuli overload. Skin-to-skin contact allows the preterm infant to benefit from the full range of “parental closeness” experiences, including the mother’s smell, touch, rhythm of movement, voice, and unique social style. This care is a particularly beneficial method because with minimal manipulation, it offers near-maximal sensory enrichment (vestibular containment) while supporting the preterm baby’s self-regulatory abilities and the organization of the biological clock with a more regular sleep-wake cycle [84], increasing weight gain and head circumference size in low birth weight preterm infants [85], and improving thermoregulation, cardiorespiratory stability, wakefulness, and behavior (less irritability) [50,86]. These authors showed that at the corrected age of three months, preterm infants were better able to manage negative emotions when confronted with unpleasant stimuli and were more successful in modulating their responses to environmental demands. At six months of age, children who had multiple skin-to-skin sessions were able to engage in more sustained object exploration during play with their mothers and demonstrated longer periods of shared attention with the mother. Parents who had skin-to-skin time described feeling helpful, feeling more familiar with their child, and having a better perception of their parenting role [87]. Mothers were more likely to breastfeed and continued breastfeeding longer [88,89]. Mothers in a kangaroo program reported better adjustment to the birth of a sick, preterm baby [90]. In general, mothers preferred this kangaroo method and were more satisfied with it than with regular incubator care [91]. The results of Feldman’s study [86] support the hypothesis that kangaroo care strengthens a mother’s attachment to her child, her sensitivity to mother–infant interactions, and her ability to decipher the newborn’s cues and respond more appropriately. The benefits of multi-sensory care have been demonstrated by various studies for over twenty years. Thus, White-Traut [58] proposed a multi-sensorial intervention including kinesthetic and vestibular solicitations, by massages, with social interactions made by the parents. This author showed an increase in the states of vigilance in preterm babies after these multi-modal stimulations. This ATVV program (Auditory, Tactile, Vestibular, and Visual stimuli) carried out by the parents offers an auditory stimulation by a soothing maternal voice associated with a tactile stimulation by massage to which is added a proposal of horizontal rocking. The intervention lasts 15 min and focuses on stimulating sensory systems developed in preterm babies. The ATVV intervention showed that multi-sensory stimulation reduced hospitalization time by increasing alertness and reducing stress states in these preterm infants: they were more responsive and showed fewer signs of disorganization. These preterm infants in the intervention group took an average of 12 days to transition from tube feeding to full self-feeding, compared to 16 days for the control group. Multi-sensory stimulation, for infants between 23 and 31 days of age, accelerated the transition to autonomous feeding with a better organization of sucking [92]. Benefits were also found in mothers with a faster decline in depressive symptoms and less parental stress. 

White-Traut, Wink, Minehart, and Holditch-Davis [93] compared this ATVV intervention with the kangaroo method in preterm infants from 31 weeks of age with respect to so-called social engagement and disengagement behaviors. The signs of disengagement, such as crying and looking away, underline, for these authors, a better learning of the regulation of the interaction: the child learns to engage and then to disengage in order to return to the interaction. Thus, the ATVV elicited more engagement (through glances towards the mother, open hands, and smiles) more subtle disengagements (gaze detour and narrowing eyes), and more marked disengagements (crying, extension patterns, and head detour) than the kangaroo care. The authors conclude that the ATVV intervention can promote the learning of the regulation of engagement and disengagement in the preterm infant. 

Gabis’ intervention program [94] also includes parents: it is designed to be integrative by reinforcing positive parent–child interactions. Parents have been guided and trained to provide sensory-motor stimulation and to assess their child’s behavioral responses in order to better support and enhance the preterm infant’s self-regulation. This approach encompasses all care procedures and refers to a range of strategies to reduce stressors in preterm infants. All the data from the study point to the crucial effect of the parental intervention on the child’s development and parental well-being. Children in the intervention group showed significantly fewer sensory difficulties, particularly in vestibular and tactile responses. The guided parenting intervention decreased levels of parental stress. Results showed improvement in cognitive, motor, and developmental skills at six months of CA and motor and language skills at two and three years of age as measured by the Bayley Scale.

The Family Nurture Intervention (FNI) program is also a multi-sensory care by parents for preterm newborns born between 26 and 34 weeks of gestational age [22,23,24]. From the first days of hospitalization, when preterm newborns are in the incubator, the mother is offered to establish contact with her child through a scented cloth, worn one night in her bra before being placed under the child’s head the next day. Mothers were encouraged to exchange these cloths at each NICU visit. The nursery nurses then helped the mothers in the FNI group make contact with their children through the incubator windows, using constant, frank, and sustained touch, speaking intonationally to their children in their native language, and, if possible, making eye contact. As soon as the preterm baby’s condition becomes more stable, skin-to-skin contact is recommended. Fathers and grandparents were also encouraged to do skin-to-skin. Then, family support sessions were set up to reassure the mother and support her for the return home. On average, the FNI mothers engaged in these return-home support activities for about 6 h per week, which was not the case for the control group mothers. Family members were also able to participate in discussion sessions to demonstrate the importance of these solicitations. The basis of the FNI is to advocate and establish reassuring routines for the child and parent within the service. This establishment of habits is based on multi-sensorial solicitations with olfaction, touch, gaze, and vocalizations of the parent in order to engage reactions and behaviors of pleasure and to support the reciprocal interactions of the dyad [23]. These habits incorporate co-regulation between mother and child and facilitate the parent–child emotional connection through the establishment of this adaptive interactive ritual between mother and child described as “The Calming Cycle”. During this calming cycle, mothers and preterm infants first go through the discomfort and distress induced by parent–infant separation, then the distress is mutually shared, then there is mutual resolution of the discomfort and distress, and finally the mutual calming may allow for periods of parent–infant interaction or sleep. Interactions between calming cycles of emotional and physiological co-regulation between mother and child result in more rapid reductions in absolute levels of discomfort and distress in both partners of the dyad [22,23,24]. Although there was no significant difference in length of hospital stay, this study showed improvement in the overall neurodevelopment and emotional development of the children in the FNI group. At term, preterm newborns in the FNI group showed changes in frontal brain activity based on electroencephalogram that are consistent with advanced maturation. At 18 months and 3 years of age, studies by Welch et al. [22,23,24] describe improved neurodevelopmental, language, and motor outcomes. Thus, repeated maternal engagement in soothing and regulating activities improves the preterm infant’s developmental trajectory and mother–infant interactions. The results of this study add further evidence to the conclusion that interventions based on parental involvement with educational support in the NICU positively improve the developmental trajectory of the preterm infant.

However, Aita’s meta-analysis [41] modulates the benefits of these multi-sensory interventions. For studies identified between 2002 and 2017, the overall quality of evidence was rated as low to very low with sample sizes that were too small. Comparing early and varied intervention programs is particularly challenging because they measure different aspects of neurodevelopment [41]. Only 12 studies met the inclusion criteria, given the nature of the interventions or the assessment instruments chosen. In addition, the time lag between the implementation of practices and their evaluation several months later (12 or 18 months) raises the question of confounding factors that may influence the results. There is a lack of longitudinal studies and clinical trials to further analyze the role of these devices [19,41] and the difference in care across NICUs [76].

### 3.3. The Difficulties of Involving Parents in the NICU 

Multi-sensory intervention programs emphasize the beneficial effects on the development of the preterm infant and on the establishment of early interactions and the first bonds of attachment. However, welcoming a preterm infant generates anxiety in the parents. Their state of stress can sometimes hinder the development of these interactions and communication skills with their preterm infant [95]. Parents feel helpless, incompetent, and unable to find the resources to accommodate their prematurely born child in the daily life of the NICU. They frequently talk about their fear of doing the wrong thing or of disturbing their own child. Therefore, despite the desire to touch or hold their child, they hold back. They only allow themselves gestures through the portholes, fingertips, and touches on the extremities of the body. Mothers have a lower evaluation of their maternal skills and may feel dispossessed of their maternal function in the face of the over-competence of the caregivers and the medical world. The place of parents within the service is also undermined by the primary medical concerns [96]. 

The parents’ initial involvement in the care process gave rise to contradictory emotions, such as joy but also stress and even anxiety [97]. Most of the parents evoked a feeling of clumsiness and a fear of harming the fragile child, all the more so if the preterm infant was less than 28 weeks of amenorrhea.

Although the parents are solicited for care in the incubator, such as toileting or diaper changes, this daily care requires mobilizations that can be disorganizing for the child, such as postural imbalances, disorganized motor skills with an increase in extensional movements, jerks and tremors, and expressions of discomfort, tonic contractions, grimaces, alert eyes, as well as vegetative reactions, such as desaturation or hiccups [98]. The performance of routine care by the parents, while allowing them to be more quickly autonomous, does not necessarily open up a time of encounter with their child. Indeed, the emotional state of the parents disturbs the decoding of the tonic manifestations of preterm babies. In the longer term, maternal (parental) stress remains higher than that experienced by mothers of full-term children until the child reaches the age of three [99]. Fathers are also at greater risk for anxiety symptoms immediately after the birth of their preterm infant [100].

Parents of preterm infants have more difficulty recognizing and adjusting to their preterm child’s cues, with less synchronous interactions and an impoverished interactive style: less touching, vocalizing, and looking [7,50]. To test the influence of this psychological state on the quality of interactions, Feldman and Granat [101] distinguished anxiety from depression during playtime. The parent with a depressed state becomes less communicative, inhibited with fewer shared glances, fewer gestures towards his or her preterm infant, and/or an increase in the duration of silences. He offers poor stimulation below the processing capacities of his baby. This parental psychological state results in a decrease in interactions. These parameters can lead to a type of insecure attachment. Concerns about the child’s well-being during skin-to-skin contact, such as the fear that the baby may stop breathing or that he or she may slip, can contribute to certain resistances and undermine the shared pleasure and interactions during skin-to-skin contact. The units, often limited in space, do not always have a parent–family–child room, and parents do not allow themselves to spend several hours in a row in skin-to-skin contact when most of them would have liked to do so. These parents may complain about a lack of adequate supervision or a lack of privacy to practice skin-to-skin. Parents often remain dependent on caregivers to initiate and terminate care. In some facilities, they cannot alert the caregivers to their desire to interrupt, which can lead to stress and radically alter the experience. Skin-to-skin in French units is proposed for 20% of parents after seven days of life, and 10% of parents did not feel confident [97]. Modifications to the skin-to-skin setup were relevant since a diagonal bending position with an adapted sling allows more visual exchanges between the mother and the preterm infant, thus improving communication behaviors [102]. Proper positioning of the preterm infant in the NICU improves self-regulation [103]. Involving mothers in the positioning of their preterm infant reduces maternal stress, promoting optimal positioning practices for the preterm infant. This is primarily a matter of supporting mothers to interact with their preterm infant sensitively, implying that they give importance to needs assessment, cues of stress, and stability of the child [103].

There are multi-sensory interventions that focus on building early relationships between parents and preterm infants aimed at harmonious parenting [104,105]. For example, the meta-analysis by Benzies [59] concludes that multi-sensory interventions involving parents result in a decrease in maternal stress and anxiety, an improvement in sensitivity, and a decrease in depressive symptoms.

## 4. Conclusions

The involvement of parents during care is a major recommendation to be valued, and the quality of this parental involvement is one of the factors that supports the neurobehavioral development of the preterm baby [106,107]. Prior training of parents by a therapist (pediatricians, nursery nurses, psychomotor therapists, or psychologists) helps to limit their concerns by teaching them the appropriate gestures, to support the implementation of care routines, and more generally to promote the sensory-motor development of the preterm baby by providing educational aids. Teaching parents to recognize and interpret the signs of their preterm child was beneficial in 73% of the interventions [76]. Quality parental involvement in this multi-sensory care promotes better attachment [22,23,24,54,55].

Therefore, it seems useful to advocate for concrete recommendations both from the point of view of the NICU environment and the parent–preterm infant dyad while being aware that the major challenge in applying these recommendations is subtle: it lies in both qualitative and quantitative degrees, in the way they are applied, and in their duration. Caution should be exercised in the introduction and progressive provision of sensory stimulation in accordance with the perceptual developmental stages of the preterm infant and under continuous monitoring of the preterm infant’s reactionary state. These recommendations should be shared by everyone involved with preterm infants and their families in the NICU (all professions). People should be made aware of the needs of the preterm infant in order to take into account the temporal organization of technical gestures.

Thus, recommendations for the NICU environment can be considered, such as minimizing excessive exposure to noise by speaking in a low voice, wearing soft shoes, not using the top of incubators as a table surface, closing porthole doors gently, responding quickly to alarms, and limiting the use of personal radios [19,108,109,110,111]. In the visual environment of the NICU, it would be desirable to keep the preterm infant in relative darkness as often as possible, not to use lights that are too bright, and, moreover, to face the baby. In order to recreate the vestibular rhythms perceived in utero, in particular the mother’s movements, it would be beneficial to rock the preterm infant slowly, regularly, and rhythmically [53]. To mobilize the tactile sensitivity of the preterm infant, it is recommended to avoid touching [4] and to prefer a frank and containing contact, carried out with the palm of the hand beforehand heated. The use of the mother’s scent is recommended to reduce the early separation imposed by premature birth. The smell of breast milk can also support the child during technical and invasive gestures [72]. The combination of horizontal vestibular lullabies and gentle humming lullabies should be studied.

This is why a new study, with preterm infant aged between 26 and 32 weeks, financed by a Hospital Program for Nursing and Paramedical Research, having obtained the authorization n° 20-0271, has been designed in order to bring together tactile, olfactory, auditory, and vestibular (often missing) stimulations. The particularity of this multicenter comparative study lies in the fact that the parents will carry out the multi-sensorial solicitations themselves after having participated in a training and practice workshop on the sensory-motor needs of the preterm newborn. This care will be performed for 20 min, once a day, for 10 days, with a gradual presentation of the solicitations, under continuous monitoring of the baby’s reactionary state. The Coding Interaction Behavior (CIB) [112] and the NNNS [113] will be performed at the end of the hospitalization to evaluate the quality of parent–child interactions and the neurodevelopment of the preterm infant.

## Data Availability

Not applicable.

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
