# Peer review of "Neonatal Care Unit Interventions on Preterm Development"

_children, 2023, doi:10.3390/children10060999_

Round 1
Reviewer 1 Report
Dear authors, I want to congratulate for this topic. It is a current topic nowadays, when, as you pointed out, neonatal resuscitation makes it possible for a large part of premature newborns to survive.
Some suggestions related to the article:
- the abstract seems to me too short and general
- the introduction is short, it could be expanded easily
- a marked section follows, like the introduction, also with 1. Please correct.
- the conclusions are too descriptive and extensive, should be summarized. Also, in conclusions it is not customary to cite other sources. These should be moved to discussions and the discussion section redone.
These are just small comments necessary to refine your article. I appreciated the clear and accessible style, as well as the serious documentation, citing over 100 references.
With a little effort on the part of the author team, the article could be accepted for publication.
I wish you success!
Minor editing of English language required
Author Response
Thank you for your suggestions;
- We have indeed taken the abstract and added some outcomes/benefits from single and multi-sensory interventions.
- In the introduction section, we have expanded on recent WHO data (with historical notions) and cited more recent articles (2022) focusing on mother-infant interactions in the care process.
- We are sorry, but we had difficulty understanding what you are saying about the numbering of the section after the introduction. We removed the number 1 before the introduction
- We have taken into account your remarks on the conclusion: indeed, we have inserted the remarks on the limits of skin-to-skin in the section on the benefits of multisensory interventions. We have also removed some redundancies but we have kept the 4 new references as we are trying to propose concrete recommendations within the NICU environment and we have not discussed this before.
Thanks you very much for your attention
Alexia and Isabelle
Reviewer 2 Report
The review article is excellently written. Follows the latest developments in Neonatal Care Unit Interventions on Preterm Development
Prematurity is a real public health issue since premature children are more fragile. This article aims to distinguish intervention programs according to the gradient of the type of solicitations: uni- or multi-modal, and according to the function of the person
performing these interventions. Most multi-sensory interventions are practiced by caregivers and it appears necessary to train and involve parents, the main co-regulator of the needs of the premature 1 child, in order to optimize this care and consequently the development of premature infants. Methodologically correctly set, facts covered by literature
It gives an overview of approaches in the care of a premature childThe review article is excellently written. Follows the latest developments in Neonatal Care Unit Interventions on Preterm Development
Prematurity is a real public health issue since premature children are more fragile. This article aims to distinguish intervention programs according to the gradient of the type of solicitations: uni- or multi-modal, and according to the function of the person
performing these interventions. Most multi-sensory interventions are practiced by caregivers and it appears necessary to train and involve parents, the main co-regulator of the needs of the premature 1 child, in order to optimize this care and consequently the development of premature infants. Methodologically correctly set, facts covered by literature
It gives an overview of approaches in the care of a premature childThe review article is excellently written. Follows the latest developments in Neonatal Care Unit Interventions on Preterm Development
Prematurity is a real public health issue since premature children are more fragile. This article aims to distinguish intervention programs according to the gradient of the type of solicitations: uni- or multi-modal, and according to the function of the person
performing these interventions. Most multi-sensory interventions are practiced by caregivers and it appears necessary to train and involve parents, the main co-regulator of the needs of the premature 1 child, in order to optimize this care and consequently the development of premature infants. Methodologically correctly set, facts covered by literature
It gives an overview of approaches in the care of a premature childThe review article is excellently written. Follows the latest developments in Neonatal Care Unit Interventions on Preterm Development
Prematurity is a real public health issue since premature children are more fragile. This article aims to distinguish intervention programs according to the gradient of the type of solicitations: uni- or multi-modal, and according to the function of the person
performing these interventions. Most multi-sensory interventions are practiced by caregivers and it appears necessary to train and involve parents, the main co-regulator of the needs of the premature 1 child, in order to optimize this care and consequently the development of premature infants. Methodologically correctly set, facts covered by literature
It gives an overview of approaches in the care of a premature child
Author Response
Thank you very much for your encouragement and your sensitivity to the problem of prematurity.
We are very grateful for your comments
Alexia and Isabelle
Reviewer 3 Report
Dear authors,
the work appears well written and complete.
However, I suggest two changes:
- update the references. Only 10 references out of 109 refer to studies published since 2020;
- deepen the section on pharmacological techniques for pain control and brain development. In particular, the use of melatonin is widely used in this segment of the population (Pain Pract. 2022 Feb;22(2):248-254. doi: 10.1111/papr.13069; Int J Mol Sci. 2021 Feb 7;22( 4):1671. doi: 10.3390/ijms22041671; Oxid Med Cell Longev. 2021 Nov 19;2021:6308255. doi: 10.1155/2021/6308255).
Author Response
We have taken your comments into account and we thank you for them.
- Indeed, we have mainly looked at the literature before 2020: this is why we have taken the time to add back 4 recent publications from 2020 (with one 2017 reference), in particular that of Lavallée et al. (2022) on the role of parents.
With regard to the recent publications proposed in connection with the reduction of pain via the use of melatonin in premature infants (particularly interesting references), we took the liberty of retaining only one, that of Cannavo (2022), which is more closely related to the behavioural benefits that we wish to study.
Thank you and best regards
Alexia SEASSAU and Isabelle CARCHON
Reviewer 4 Report
Dear authors,
The subject covered by your manuscript is a very important one. Since more and more premature newborns survive worldwide, and with very small gestational ages, the study of their neuropsychomotor development becomes of great importance. Every effort must be made for these newborns not only to survive, but to do so in an environment as suitable as possible for physiological development. I therefore consider that your review is welcome and deserves to be published.
Some small typographical errors need to be corrected. The Conclusions chapter should be more concise.
Author Response
Thank you very much for your encouragement and your sensitivity to the problem of prematurity. We are very grateful for your comments. On your advice, we have indeed moved and deleted data from the conclusion to make it lighter.
Thank you again and best regards
Alexia SEASSAU and Isabelle CARCHON